# Mechanical Behavior and Excavation Optimization of a Small Clear-Distance Tunnel in an Urban Super Large and Complex Underground Interchange Hub

Jianxiu Wang [1,2,*], Ansheng Cao [1], Zonghai Li [3,*], Zhao Wu [1], Lihua Lin [3], Xiaotian Liu [1], Huboqiang Li [1] and Yuanwei Sun [1]

1   College of Civil Engineering, Tongji University, Shanghai 200092, China
2   Key Laboratory of Geotechnical and Underground Engineering of Ministry of Education, Tongji University, Shanghai 200092, China
3   Xiamen Road and Bridge Construction Group Company Ltd., Xiamen 361026, China
*   Correspondence: wang_jianxiu@163.com (J.W.); lzh88623667@163.com (Z.L.);
    Tel.: +86-13916185056 or +86-21-65983036 (J.W.); +86-15960823667 (Z.L.); Fax: +86-21-65985210 (J.W.)

**Abstract:** Close excavation section spacing, the mutual influence and interpenetration of various processes, and the multiple disturbances of the middle rock pillar in a small clear-distance tunnel pose great difficulty to construction. This study adopted the small clear-distance tunnel of Xiamen Haicang Evacuate Channel as the research object. The tunnel belonged to a small clear-distance tunnel in an urban super large and complex underground interchange hub where complex adjacent small clear-distance tunnels were adopted. ABAQUS was used to analyze the influence of different excavation schemes, lithological grades, and footage lengths for tunnel stability. The deformation and stress characteristics of the tunnel's surrounding rock and lining structure in different excavation schemes (full section method, bench method, center diaphragm (CD) method, and double wall heading method), lithological grades (III, IV and V), and footage lengths (3 m, 4 m and 5 m) were introduced. The results showed that the double wall heading method could effectively control the horizontal displacement of the hance, and the overall stress state of the lining in the CD and double wall heading methods were reasonable. The vertical displacement of the surface and vault was positively correlated with the elastic modulus of the rock mass. When no temporary support was present in the grade V rock mass, the area from the hance to the arch foot was prone to large deformation. Reducing the footage was beneficial to controlling the deformation of the vault and hance. This study can provide a reference for the on-site construction of small clear-distance tunnels.

**Keywords:** small clear-distance tunnel; excavation schemes; lithological grades; footage lengths; numerical simulation



## 1. Introduction

The environmental protection requirements of tunnels have become increasingly stringent in recent years because the amount of land resources is decreasing. Small clear-distance tunnels are widely used in highway tunnels because of their unique advantages, such as minimal environmental damage, minimal land occupation, beautiful appearance, and convenient route planning [1,2]. During the construction of small clear-distance tunnels, the middle rock pillar must have sufficient strength and stability [3], and the independence and integrity of the surrounding rocks and lining systems must be maintained. Different from general separated tunnels, the disturbance of the two adjacent tunnels of a small clear-distance tunnel exerts a great impact during construction. After antecedent tunnel excavation, the initial stress conditions of the surrounding rock are changed, and the excavation of the rear tunnel is affected. The excavation of the rear tunnel can easily cause the deformation of the middle rock pillar, which is not conducive to the stability of tunnel

construction [4–6]. Therefore, the construction safety of small clear-distance tunnels cannot be ignored.

Extant research on small clear-distance tunnels at home and abroad has mainly adopted mechanical analysis, field monitoring, model testing, and numerical simulation. On the basis of complex variable theory, Fu et al. [7] studied the analytical solutions of deformation, displacement, and stress of double parallel tunnels in the elastic half-plane based on the complex theory. Considering the radial stress at the tunnel boundary, Kooi et al. [8] proposed an analytical expression for calculating the stress and displacement around two parallel deeply buried tunnels using the Schwarz alternating method. Yan et al. [9] derived the surrounding rock stress and displacement of a shallowly buried, twin-parallel tunnel in consideration of the uniform radial stress boundary condition. To study the influence of different tunnel clear distances and surface slopes on the surrounding rock pressure of complex, eccentric, shallow-buried, small, clear-distance tunnels, Li et al. [10] used the design specifications for highway tunnels, established a surrounding rock pressure calculation model, and deduced the theoretical calculation formula of tunnel surrounding rock pressure. On the basis of the actual slip surface of a tunnel, Sun et al. [11] established a load calculation model for small clear-distance tunnels and proposed a calculation method for the surrounding rock pressure of small clear-distance tunnels. In terms of model tests, Lei et al. [12] established a tunnel excavation simulation test system under typical asymmetric loading and applied this system to analyze the failure mechanism and load characteristics of the surrounding rock of a small, clear-distance, shallow-offset tunnel. Jiang et al. [13] designed a physical test model of a shallow-buried, bias-pressure, small, clear-distance tunnel with a ratio of 1:10 and conducted a large shaking table test to study the influence of seismic wave type and acceleration excitation peak value on the strain response law of the tunnel lining. In terms of field measurement, Hiroshi et al. [14] analyzed the influence of different middle rock pillar reinforcement methods on surrounding rock stability by examining the Fukuoka Metro Line 3 Project and combined on-site monitoring and measurement data. However, the field tests only focused on specific projects and lacked a systematic summary of the monitoring data. Numerical simulation has the advantages of low time consumption, low cost, and high calculation precision, and it has been widely used in the calculation of small clear-distance tunnels. Ng et al. [15] studied the influence of the excavation sequence and lag distance on the load transfer mechanism and mechanical behavior of a small clear-distance tunnel. Yao et al. [16] analyzed the influence of clear distance, buried depth, and other conditions on the blasting vibration characteristics of a small clear-distance tunnel through numerous 2D and 3D numerical simulations. For a small clear-distance tunnel, Xu et al. [17] performed a numerical analysis and compared the effects of different construction methods, such as step, advanced reinforcement, and grouting reinforcement. Yu et al. [18] applied numerical simulation combined with a model test to examine the two transition sections from the large-span section to the small clear-distance section of a bifurcated tunnel. They showed that the vault and middle rock pillar are greatly affected by tunnel excavation. On the basis of a shield tunnel project, Zhao et al. [19] established a shield tunnel construction model using refined numerical simulation technology to analyze the interaction of small clear distance and shallow-buried shield construction and investigated the interaction law of shield tunnel construction under different clear distances. Hage et al. [20,21] analyzed the influence of different layouts and excavation sequences of a double-line shield tunnel on surface settlement and the surrounding environment. Using the ANSYS finite element program, Zhang et al. [22] studied the stress and deformation characteristics of a shallow-buried, bias-pressure, small, clear-distance tunnel under different bias angles, spacings, and buried depths.

Selecting a reasonable construction method was an important factor to ensure the stability of surrounding rock and construction safety [23,24]. At present, the commonly used construction methods for tunnels included the bench method, cross diagram (CRD) method, center diaphragm (CD) method, and double wall heading method. The main idea was to convert the large section of the whole tunnel into small section construction, reduce

the span of each excavation, and ensure the safety of construction [25–28]. Many scholars had explored tunnel construction methods. Liu et al. [29] carried out model tests to restore the surrounding rock displacement and stress release laws of the CD method, double wall heading method, and bench method in the process of partial excavation and found that the surrounding rock deformation experienced a slow increase-sharp increase-stable state. As a common method of tunnel construction, the CD method had the advantage of resisting the inward horizontal convergence of the tunnel. Based on the CD method, Zhou et al. [30] put forward the vertical center diaphragm (VCD) method of axisymmetric structure, and studied the deformation law and mechanical characteristics of the VCD method through numerical calculation and field comparative tests. Wang et al. [31] used the CRD method and the four-step method to study the excavation of shallow and large-diameter twin-tunnel in soft ground. Jin et al. [32] established a three-dimensional calculation model to study the influence of tunnel excavation mode on stratum deformation. It was found that the double wall heading method can better control surface deformation and has higher construction efficiency. Xu et al. [33] built a three-dimensional numerical model of a large section tunnel by finite difference method, analyzed the construction of the double wall heading method, and discussed the deformation characteristics and stress conditions of surrounding rock and tunnel structure.

Many scholars have studied the small clear-distance tunnels. However, there are few reports on the comparison of different construction methods of small clear-distance tunnels. The stress and displacement of the ultra-small clear-distance tunnel are complex. At present, there is no unified construction specification, and a unified and persuasive theory has not been formed. In addition, the surrounding rock stress of a small clear-distance tunnel is affected by lithology and tunnel excavation mode, so the construction and support structure design of a small clear-distance tunnel face many problems. Complex adjacent tunnels form complex small clear-distance tunnels, especially for an urban super large and complex underground interchange hub where main line and ramps are all constructed underground. The construction mechanical behaviour of the small clear-distance tunnels is vital for an underground engineering system. The small clear-distance tunnel of Xiamen Haicang Evacuate-channel was selected as the background; the influence of different excavation schemes, lithologic grade, and footage length of the small clear-distance tunnel on the stability of the tunnel, the construction mechanical characteristics, and laws of the small clear-distance tunnel were obtained; and the corresponding construction countermeasures are put forward. The results can provide a reference for similar engineering projects.

## 2. Research Section Project Overview

The Xiamen Haicang Evacuate Channel Project is located in the Haicang District, Xiamen. The project has tunnels #1 and #2. Tunnel #2 and the main line tunnel of Lu'ao Road form the Lushu Interchange at the node of Xinmei Road, and a total of four ramps are arranged at the interchange in a semi-interchange form. As shown in Figure 1, in the process of gradually forming a bifurcation tunnel in tunnel #2, the bifurcation forms two ramps (A and D). The research section is the small clear-distance section of the left line of tunnel #2 and ramp A. The small clear-distance section of the main line of tunnel #2 is ZK2+665-ZK2+775, and ramp A's small clear-distance section is AK1+600-AK1+490, with a length of 110 m.

The maximum excavation span of the main tunnel section is 16.17 m, the height is 11.32 m, and the limit span is 14.45 m. The maximum excavation span of ramp A is 12.62 m, the height is 10.40 m, and the limit span is 11.10 m. The research section mainly passes through the second intrusive granite stratum, which is dominated by moderate weathering. It is distributed with grade III surrounding rocks, which are relatively complete rocks. It is located in a deeply buried tunnel in the mountains with a burial depth of nearly 100 m. For the research section, this study adopts the step method, short step method, CD method, and double wall heading method for a comparative analysis. The face of the main line and ramp is divided into several parts for step-by-step excavation, and the distribution of

the excavation is shown in Figure 2. The numbers in the Figure 2 represent the different rock mass part. The excavated soil in each block is divided by the primary support on the contour and the temporary support inside the section, and a composite lining is applied. The composite lining consists of a primary bolt-shotcrete steel mesh support, a temporary I-steel frame support, and a secondary lining concrete layer of the arch wall and inverted arch. The parameters of the primary support, secondary support, and temporary support in the research section are shown in Table 1.

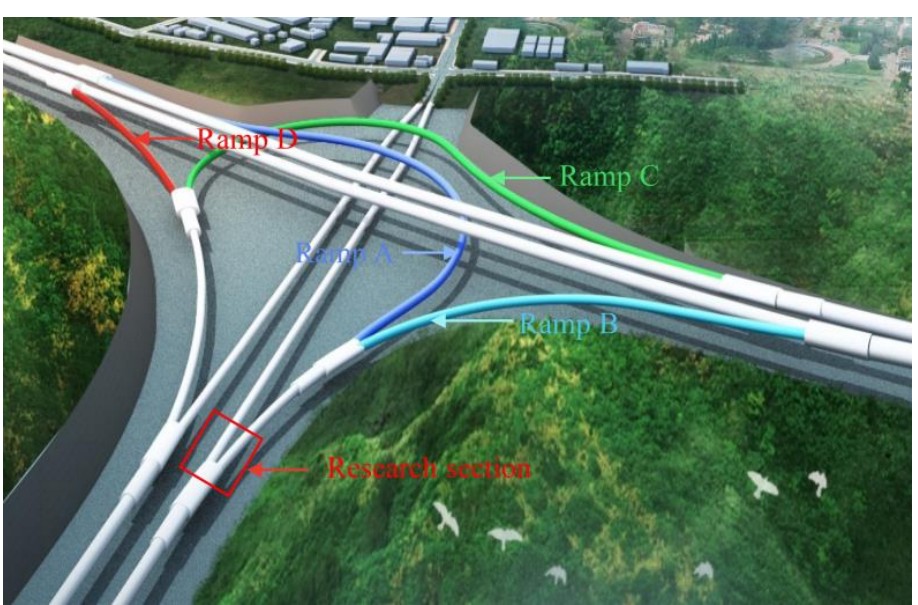

**Figure 1.** Schematic diagram of the underground interchange hub including the evacuate channel and Luao Road.

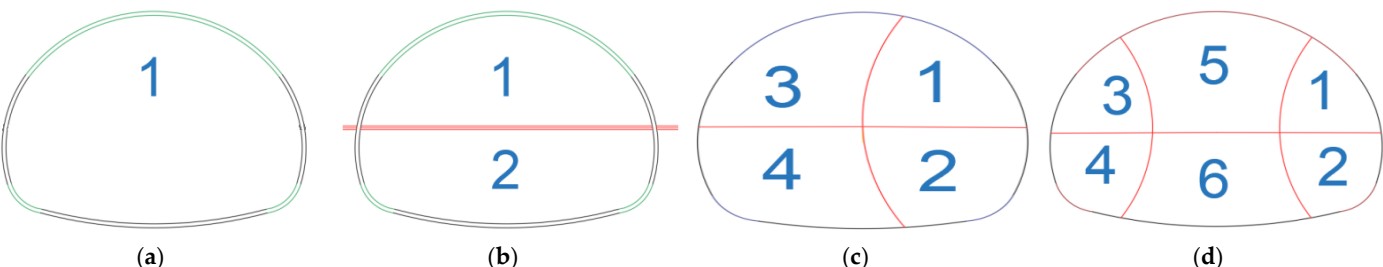

(a)   (b)   (c)   (d)

**Figure 2.** Face division of each excavation scheme: (**a**) Full section method; (**b**) Step method; (**c**) CD method; (**d**) Double wall heading method.

**Table 1.** Research section support parameters.

| Name of the Supporting | | Support Method |
|---|---|---|
| Main tunnel | | |
| Primary lining support | Concrete spraying layer<br>Rock bolt<br>Reinforcing mesh | C25 shotcrete 20 cm<br>Φ22 mortar bolt L300 cm<br>Φ8 reinforcing mesh 25 cm × 25 cm |
| Temporary support | I-steel truss<br>Concrete spray layer<br>Reinforcing mesh | I18 I steel frame, longitudinal spacing 50 cm<br>C25 concrete spray layer 22 cm<br>Φ8 reinforcing mesh 20 cm × 20 cm |
| Secondary lining support | Arch wall<br>Inverted arch | C30 Waterproof reinforced concrete 40 cm thick<br>C30 Waterproof reinforced concrete 40 cm thick |

**Table 1.** *Cont.*

| Name of the Supporting | | Support Method |
|---|---|---|
| Ramp section | | |
| Primary lining support | Concrete spraying layer<br>Rock bolt<br>Reinforcing mesh | C25 shotcrete 20 cm<br>Φ22 mortar bolt L2500 cm<br>Φ8 reinforcing mesh 25 cm × 25 cm |
| Temporary support | I-steel truss<br>Concrete spray layer<br>Reinforcing mesh | I18 I steel frame, longitudinal spacing 50 cm<br>C25 concrete spray layer 22 cm<br>Φ8 reinforcing mesh 20 cm × 20 cm |
| Secondary lining support | Arch wall<br>Inverted arch | C30 Waterproof reinforced concrete 40 cm thick<br>C30 Waterproof reinforced concrete 40 cm thick |

## 3. Numerical Model

### 3.1. Establishment of a 3D Numerical Model

Given that the influence range of the excavation stress was 3–5 times that of the tunnel radius, the boundary dimensions of the model were 200 m in the X direction, 110 m in the Z direction, and 160 m in the Y direction. The buried depth of the tunnel was 100 and 60 m away from the bottom boundary. The numerical model was established using the software Abaqus. The stratum, tunnel, primary lining, secondary lining, and temporary support were simulated using the eight-node reduced integration solid element which was defined as C3D8R in Abaqus. The thickness of the primary lining solid unit was 20 cm, the temporary support was mainly carried by I-steel, the thickness of the solid unit was 18 cm, and the thickness of the secondary lining solid unit was 40 cm. The two tunnels were mutually angled, the main section was along the Z-axis direction, and the ramp section was along the Z-axis horizontal with an angle of 8.28°. The minimum clear distance in the model was 1.6 m. The two tunnels, from a small clean distance, gradually transition to a separate tunnel. The model was divided by a hexahedral mesh, and local mesh encryption was carried out for important positions, such as the tunnel excavation section and surrounding rock. The model was divided by a hexahedral grid, and the local grid of the tunnel excavation section was densified. With the double wall heading method and CD method as examples, the overall numerical model was built and is shown in Figure 3a. A total of 117,355 units were generated by the model, among which the solid element grids, such as excavated rock mass, primary lining, secondary lining, and temporary support, are shown in Figure 3b. The front, back, two sides, and bottom surfaces of the model were constrained by normal displacement, and the front excavation face and top surface were not constrained.

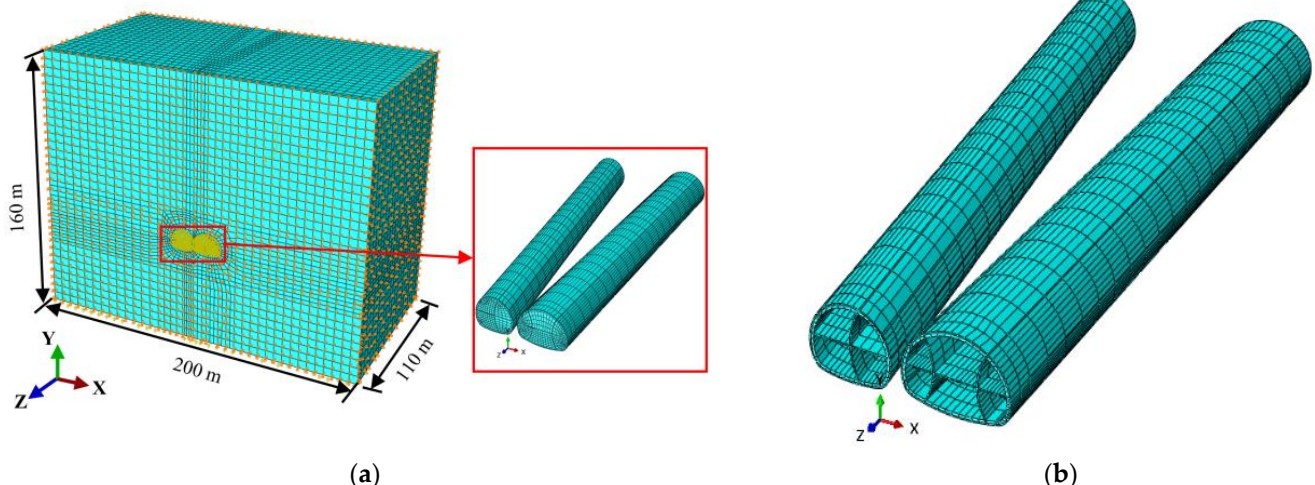

(**a**)                    (**b**)

**Figure 3.** Numerical calculation model: (**a**) Overall numerical model; (**b**) Tunnel lining.

### 3.2. Numerical Model Parameters

The equivalent continuum method was used to represent the surrounding rocks and linings in the numerical simulations. According to the survey report, the rock mass in the research section was grade III. The physical and mechanical parameters of the surrounding rock mass and lining in the numerical simulation were determined and adjusted by referring to the Specifications for Design of Highway Tunnels [34]. As an elastoplastic body, the rock mass conformed to the Mohr-Coulomb failure criterion, and the lining structure was an elastomer. The material parameters in the numerical simulation are shown in Table 2.

**Table 2.** Material parameters of the numerical model.

| Materials | Density (kg/m$^3$) | Elastic Modulus (GPa) | Poisson's Ratio | Cohesion (MPa) | Friction Angle (°) |
|---|---|---|---|---|---|
| Surrounding rock | 2200–2300 | 6–10 | 0.25–0.3 | 2.0 | 39–50 |
| Primary lining support | 2500 | 30 | 0.20 | / | / |
| Secondary lining support | 2500 | 32.5 | 0.15 | / | / |
| Temporary support | 7900 | 200 | 0.30 | / | / |

## 4. Numerical Results and Discussion

### 4.1. Stability Analysis of Small Clear-Distance Tunnels under Different Excavation Schemes

Different excavation schemes were selected when tunnelling was carried out under different surrounding rock levels and buried depths. Given that the project was a small clear-distance tunnel and the minimum clear distance in the model was 1.6 m, to ensure the stability of the construction interruption surface, the excavation scheme needed to be optimized and analyzed. Therefore, commonly used excavation schemes, such as full section, step, CD, and double wall heading methods were simulated. Figure 4 shows the layout of each excavation scheme. In each excavation scheme, the main line of the tunnel was excavated first with footage of 5 m, and the staggering distance between the main line and the ramp was 15 m. The section 20 m away from the main line tunnel portal was selected as the target section for analysis.

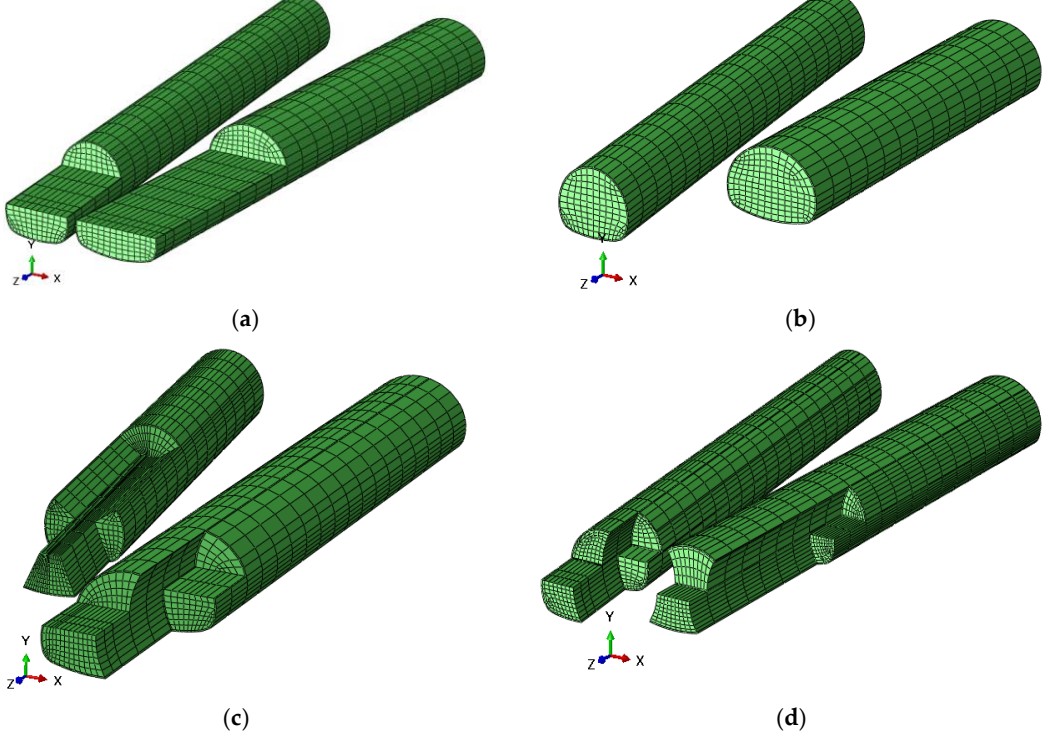

**Figure 4.** The layout of each excavation scheme: (**a**) Step method; (**b**) Full section method; (**c**) CD method; (**d**) Double wall heading method.

### 4.1.1. Analysis of Surface and Surrounding Rock Deformation

The ramp excavated to the target section was selected, and the support time for analysis was completed. Figure 5 shows the vertical displacement of the surrounding rocks in the various excavation methods. The overall vertical displacements of the surrounding rocks were close. Given that the right tunnel was excavated first and the left ramp was excavated later, the overall vertical displacement exhibited uneven settlement on the same horizontal plane and eventually settled to the antecedent tunnel. The different excavation schemes had different disturbance methods and times relative to the surrounding rocks, resulting in different vertical displacement trends of the surrounding rocks. The comparison of the vertical displacements of the surrounding rocks above the ramp showed that the order of influence range from small to large was as follows: full section method < CD method and double wall heading method < step method. The excavation of full section method was completed in one step, the influence range of which on vertical displacement was less than the methods using multi-step. Although the CD and double wall heading methods provided more disturbances to the surrounding rocks than the other methods did, the ground deformation did not extend to the vault due to the vertical bearing of the temporary support.

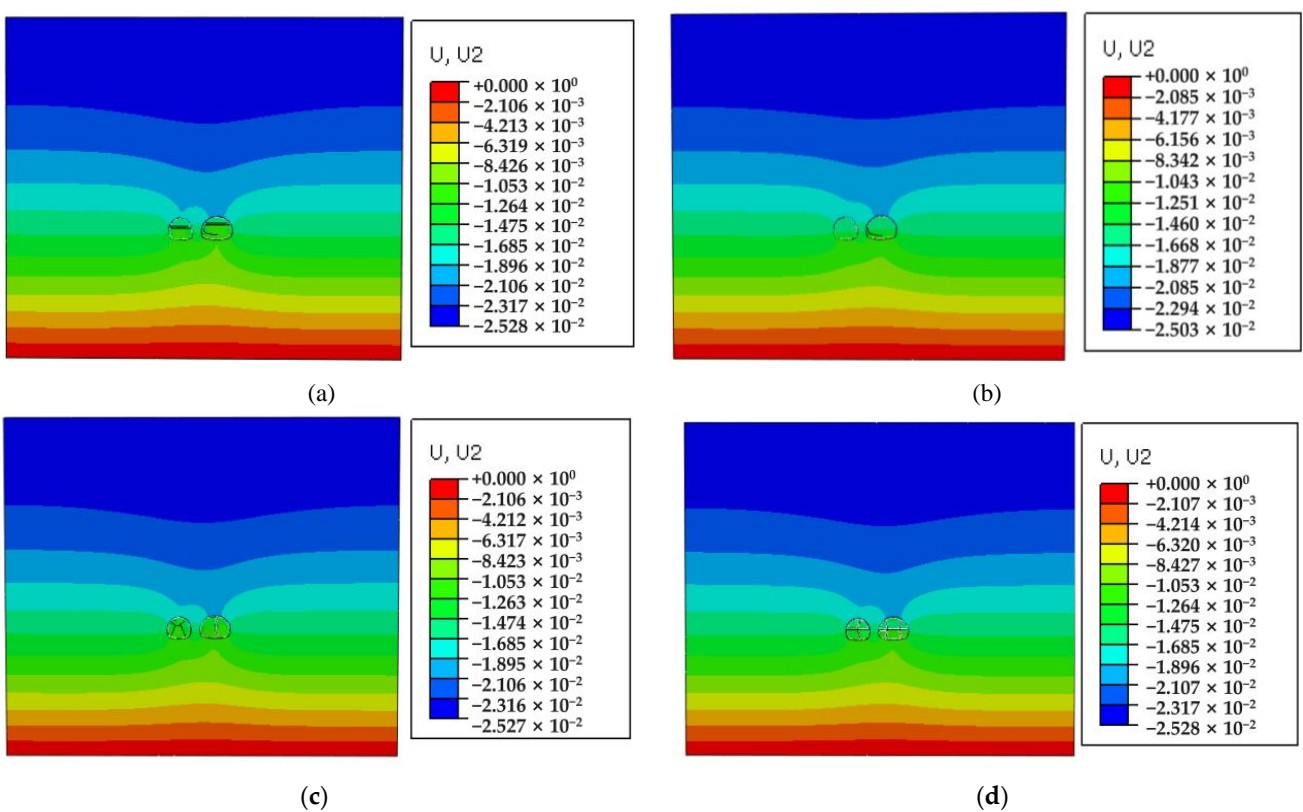

**Figure 5.** Overall vertical displacement of surrounding rock (unit: m): (**a**) Step method; (**b**) Full section method; (**c**) CD method; (**d**) Double wall heading method.

Figure 6 shows the horizontal displacement of the surrounding rocks under the various excavation methods. The temporary support exerted an obvious effect on controlling the horizontal displacement. Under the step and full section methods, due to the large excavation range at one time, the surrounding rocks around the tunnel had a large range of horizontal displacement, which easily produced stress concentration on the lining that was not conducive to the overall stress of the lining structure. In the CD and double wall heading methods, the overall horizontal displacement was relatively average, which helped improve the integrity of the lining structure and the stability of the overall displacement of the surrounding rocks.

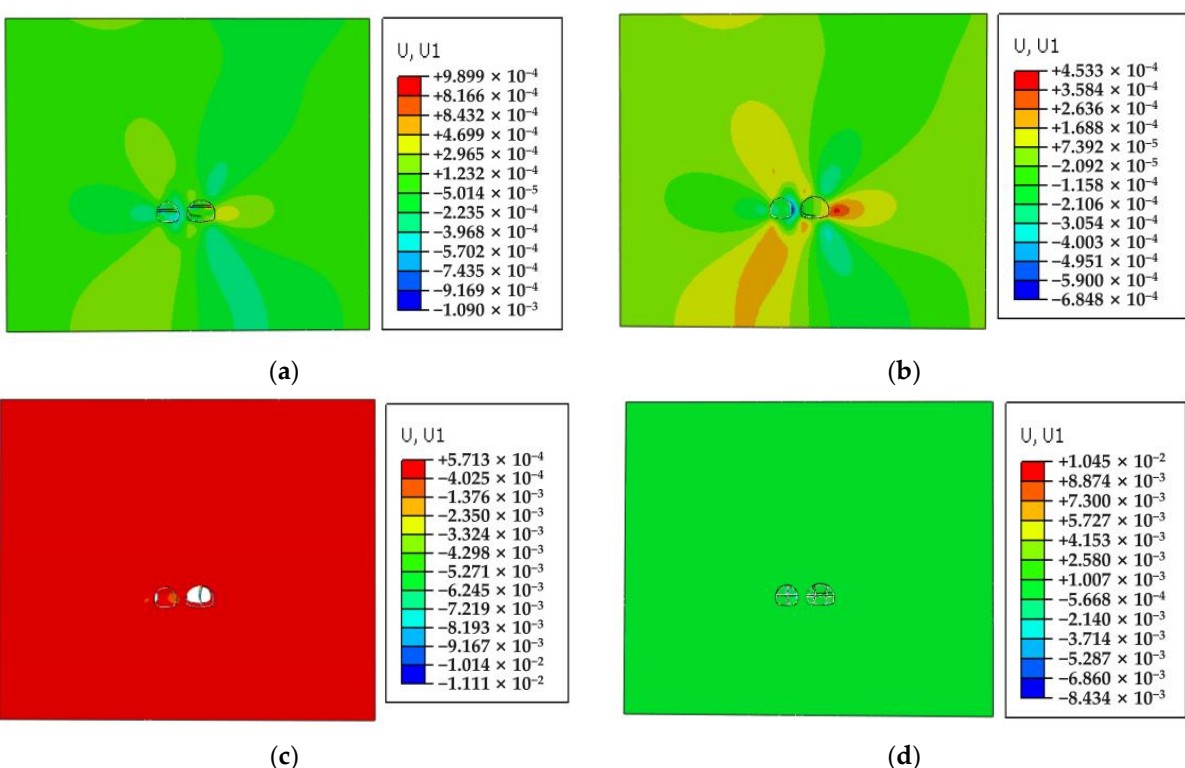

**Figure 6.** Overall horizontal displacement of surrounding rock (unit: m): (**a**) Step method; (**b**) Full section method; (**c**) CD method; (**d**) Double wall heading method.

### 4.1.2. Deformation Analysis of the Surrounding Rock and Lining

The target section was selected to analyze the difference in vault deformation between the main line and ramp. Figure 7 shows the vertical displacement variation curve of the main line and ramp vault with the tunnelling process. A rapid decline followed by levelling was observed in the vertical deformation of the tunnel vault, which corresponded to the complete excavation of the main line and ramp in the target section. In terms of displacement, the order was CD method < double wall heading method < step method = full section method, indicating that compared with the full section method, although the CD method increased the construction step sequence, the initial settlement occurred slowly, and the absolute settlement was lower. The double wall heading method could maintain a platform with long construction steps before the curve drop. The comparison of the settlement difference between the main tunnel and the ramp at the vault revealed that the deformation of the main tunnel excavated first was greater than that of the ramp.

Figure 8 shows the horizontal displacement of the lining of the main line and ramp under each excavation scheme after the tunnel was penetrated. Figure 8a,b indicate that the horizontal displacement under the step and full section methods was mainly concentrated at the inner hance of the two tunnels. The displacement at the front of the tunnel was large. With the gradual construction of the lining, the horizontal displacement at the rear of the tunnel decreased gradually. Figure 8c,d show that in the schemes with temporary support, such as the CD method and double wall heading methods, the overall horizontal displacement of the lining structure was mainly concentrated on the temporary structural deformation, and the distribution on the primary lining was uniform, which was conducive to the overall stability.

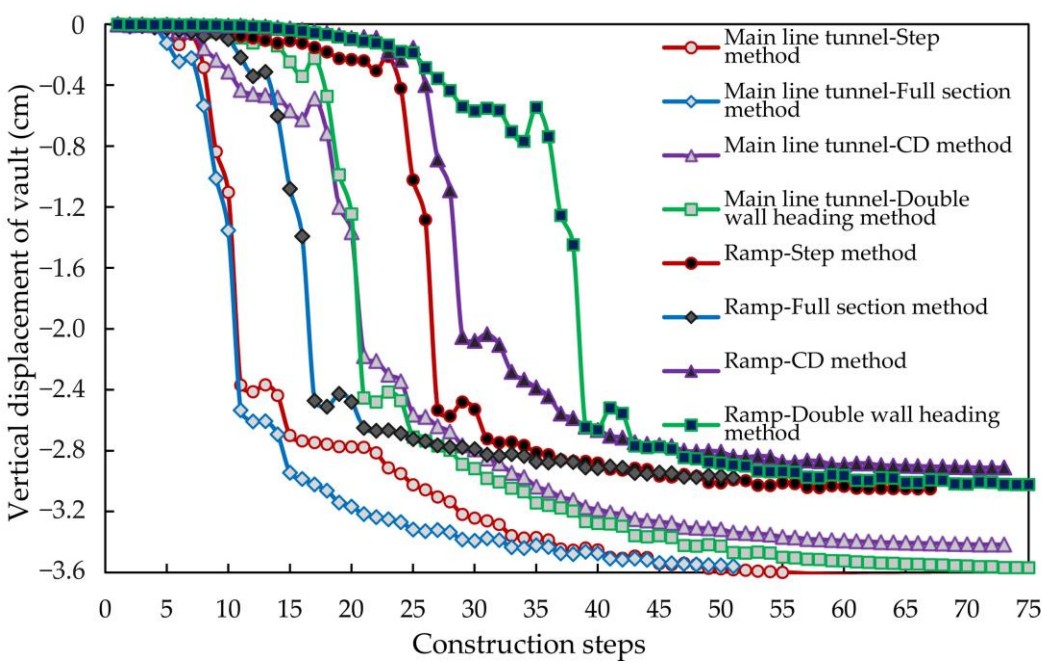

**Figure 7.** Variation curve of vertical displacement of the vault in the small clear-distance tunnel.

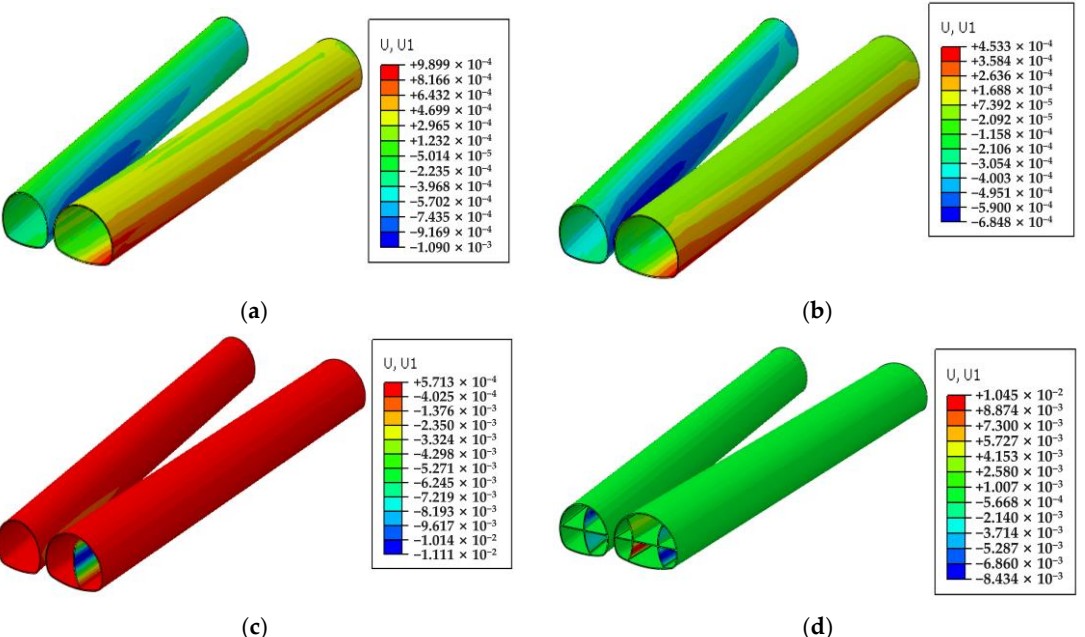

**Figure 8.** Horizontal displacement of the lining (unit: m): (**a**) Step method; (**b**) Full section method; (**c**) CD method; (**d**) Double wall heading method.

The monitoring points in the target section marked in Figure 9 were selected to measure the deformation of the hance of the two tunnels. Figure 10 presents the horizontal displacement curves of the monitoring points along the main tunnel and ramp as the construction progressed. The horizontal displacement curve of the main line tunnel in the step, CD, and double wall heading methods initially declined and then rose. The analysis showed that when the excavation reached the section near the monitoring point of the main line tunnel, the rock mass near the hance could not be fully excavated to release the stress, and the hance was squeezed during the deformation of the lower rock mass. With the excavation and lining of the lower rock mass, the hance displacement could be controlled after forming a closed loop. The section could be quickly closed due to the one-time

excavation and support, and the fluctuation of the horizontal displacement of the hance was low. Given that the double wall heading method involved the temporary support in the horizontal direction, it had the best control effect on the horizontal displacement, and the final displacement was close to 0. The ramp horizontal displacement curve presented a downward trend with the construction step; that is, a trend of left horizontal displacement was observed at the hance. In terms of the final horizontal displacement, the order was double wall heading method < CD method < step method = full section method.

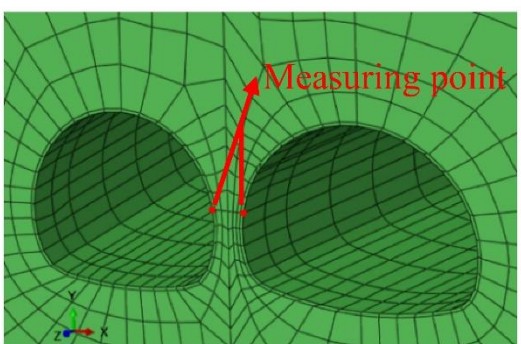

**Figure 9.** Layout of monitoring points of hance in the small clear-distance tunnel.

**Figure 10.** Horizontal displacement of hance in small clear distance section: (**a**) Step method; (**b**) Full section method; (**c**) CD method; (**d**) Double wall heading method.

Generally, the double wall heading method could maintain the integrity of the primary lining displacement, and the horizontal displacement of the primary lining at the hance was small. No violent fluctuation occurred due to the effect of lateral support in the deformation process, which showed the effectiveness of lateral temporary support in controlling hance deformation.

### 4.1.3. Stress Analysis of the Supporting Structure

Given that the primary lining was mainly a concrete material, which has high compressive capacity but low tensile capacity, the main tensile parts can be determined by the overall stress distribution on the supporting structure. Figure 11 presents the maximum principal stress distribution of the main line tunnel and ramp support structure after the tunnel was penetrated using the different excavation schemes. Positive values in the figure indicate tension, and negative values indicate compression.

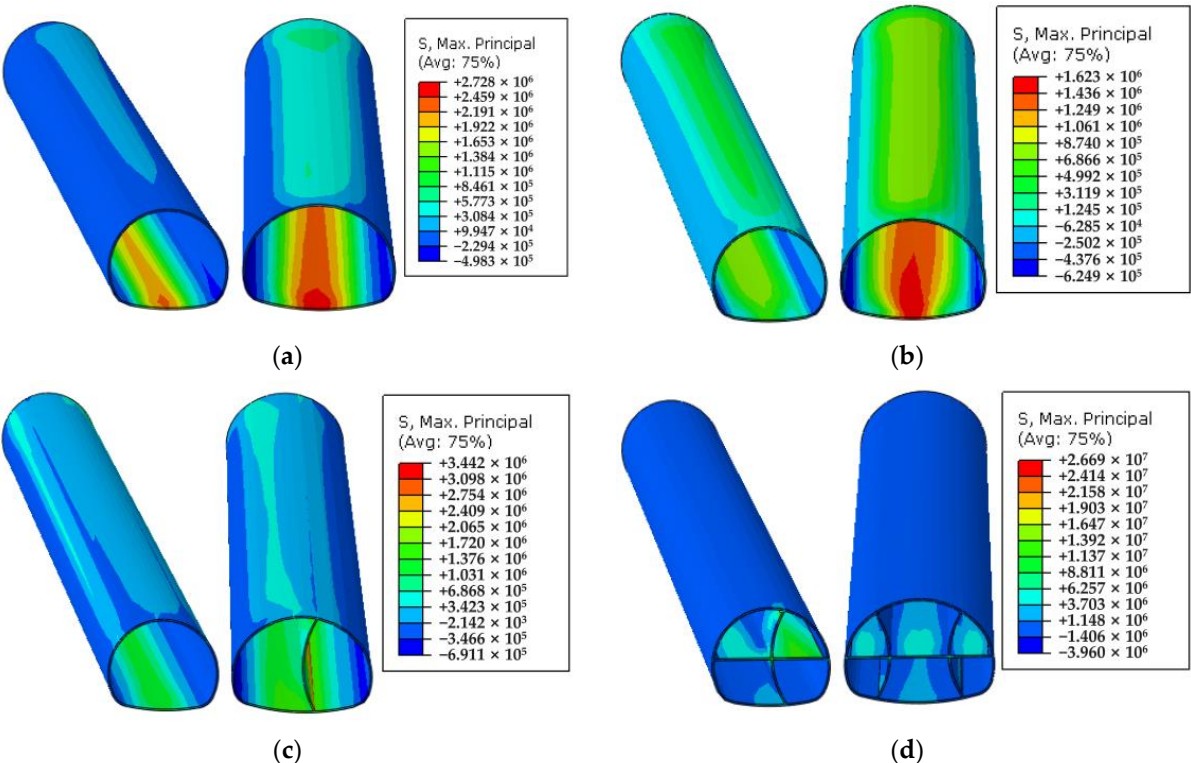

**Figure 11.** Maximum principal stress of lining structure (unit: Pa): (**a**) Step method; (**b**) Full section method; (**c**) CD method; (**d**) Double wall heading method.

Figure 11 reveals that tensile state distribution existed around the arch in the step and full section methods without temporary support. In the full section method, the main tunnel and ramp were almost distributed in length, which was not conducive to the bearing performance of the primary lining structure. Figure 11 also shows that tension stress existed around the vault without temporary support, such as in the step and full section methods. In the full section method, the main line tunnel and ramp were almost distributed throughout the length, which was not conducive to the pressure bearing performance of the primary lining structure. Local tensile stress was observed in the CD method, but almost no tensile stress was observed in the double wall heading method. In the full section and step methods, the maximum principal stress was distributed in the inverted arch. Meanwhile, the inverted arch of the CD and double wall heading methods could effectively bear the load transmitted from the upper lining to the lower part without a large maximum principal stress. The distribution range and value of the principal stress of the ramp invert were smaller than those of the main line tunnel, which was highly obvious in the step and full section methods. The reason was that after the excavation of the main line, the stress

was first released on the lining of the main line tunnel. However, if the main line tunnel was equipped with temporary support, the disadvantageous stress position of the main tunnel could be effectively adjusted, as shown in Figure 11c,d.

Generally, the four excavation schemes had little influence on the vertical deformation of the vault. The CD method had the smallest vertical displacement, and the double wall heading method effectively controlled the horizontal displacement of the hance, which was similar to the research conclusions of scholars in large section tunnels [35,36]. The overall stress of the lining was reasonable in the CD and double wall heading methods with temporary support, but a large range of tensile stress was observed in the full-section and step methods.

### 4.2. Analysis of the Construction Behaviour of the Small Clear-Distance Tunnel under Different Lithology Grades

Grades III–V surrounding rocks were distributed in the range of ZK2+780–ZK3+120 of the main line tunnel. Therefore, a simulation of small clear-distance tunnel construction with grades III–V surrounding rocks was conducted to determine the possible dangerous parts in advance. The other condition parameters in the model remained constant, and the grid division was the same. The tunnel was 110 m long, and the step method was adopted. The main line tunnel was excavated first, followed by the ramp. The staggering distance between the main line and the ramp was 25 m, and the staggering distance between the bench face and the tunnel face was 25 m. The section 20 m away from the main line tunnel portal was selected as the target section for analysis. The parameters of three types of the surrounding rock are shown in Table 3.

**Table 3.** Parameters of three types of the surrounding rock.

| Materials | Density (kg/m³) | Elastic Modulus (GPa) | Poisson's Ratio | Cohesion (MPa) | Friction Angle (°) |
|---|---|---|---|---|---|
| Grade III rock mass | 2200 | 6 | 0.25 | 2.1 | 50 |
| Grade IV rock mass | 1900 | 3 | 0.30 | 1.4 | 30 |
| Grade V rock mass | 1600 | 1.5 | 0.35 | 0.7 | 10 |

#### 4.2.1. Deformation Analysis of the Surrounding Rocks and Lining

Figure 12 shows the vertical displacement curves of the ground, main line, and ramp vault with different grades of surrounding rocks in the target section as the construction progressed. The surface settlement revealed that the worse the rock mass quality was and the lower the grade was, the greater the surface settlement was. The final surface settlement of grades III, IV, and V was 1.71, 2.70, and 4.73 cm, respectively, and the ratio was basically stable at 1.5–1.7. Meanwhile, the ratio of elastic modulus in the lithologic parameters was constant at 2, and the elastic modulus was the controlling value of surface settlement; the two showed a positive correlation. The vertical displacement of the vault of the main line and ramp lining had a falling section at the same construction step, which corresponded to the vault settlement caused by the excavation of the tunnel face.

Table 4 shows the relationship among the surface, main line, and ramp vault displacements under different lithology grades. With the deterioration of lithology quality, the displacement of the surface and vault increased. The difference between the mean values of surface and vault settlement increased with the deterioration of the lithologic grade. Surface settlement accounted for about 8% of the vertical displacement of the vault. The proportion increased with the deterioration of the lithologic grade.

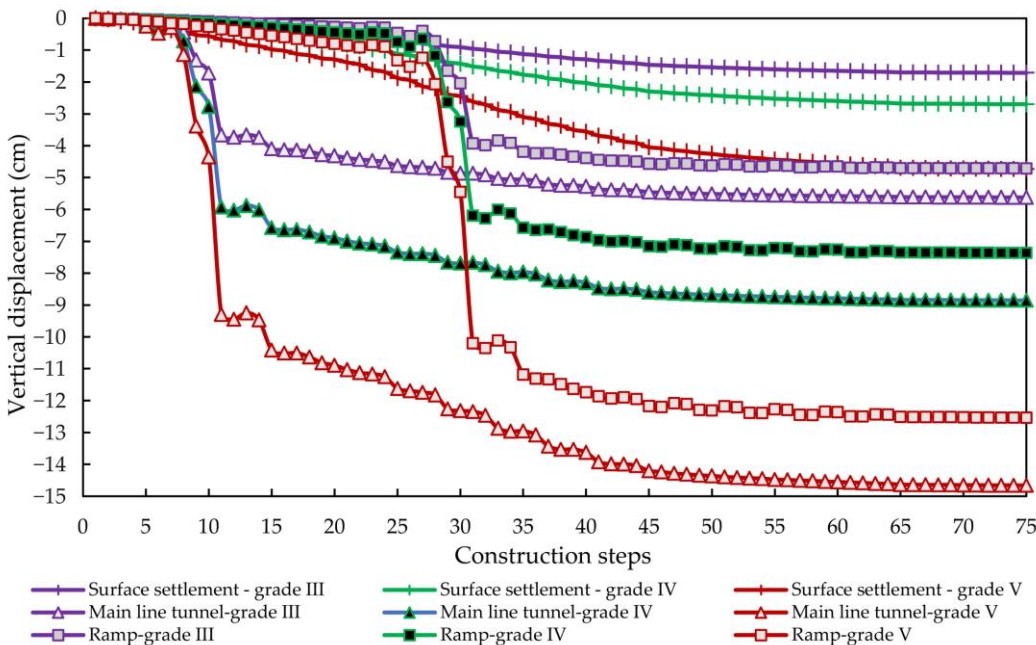

**Figure 12.** Vertical displacement curves of the ground and vault with different grades of the surrounding rock.

**Table 4.** Vertical displacement under various lithologic grades.

| Lithologic Grades | Ground Settlement (cm) | Displacement of Main Tunnel Vault (cm) | Ramp Vault Displacement (cm) | Mean Difference between Surface and Vault Settlement (cm) | Ground Settlement/Vault Displacement |
|---|---|---|---|---|---|
| III | 1.71 | 5.62 | 4.71 | +3.45 | 8.28% |
| IV | 2.70 | 8.85 | 7.36 | +5.41 | 8.31% |
| V | 4.73 | 14.66 | 12.54 | +8.87 | 8.69% |

Figure 13 shows the horizontal displacement distribution of the surrounding rocks and lining after tunnel penetration. The large horizontal displacement in the surrounding rocks of grades III and IV was mainly distributed from the hance to the arch foot at both ends of the tunnel, and the displacement of the surrounding rocks was driven by the horizontal deformation at both ends of the tunnel. Comparison of the horizontal displacements of the surrounding rocks in grades III and IV revealed that the horizontal displacements of the grade IV surrounding rocks had a wide distribution range, whereas in the grade V surrounding rocks, although the overall displacement was relatively uniform, the absolute value of the displacement was the largest. The horizontal displacement of the lining indicates that the maximum horizontal displacement on the lining was at the hance. Therefore, the displacement of the hance needed to be analyzed further. The hance monitoring point in Figure 9 was selected to analyze the horizontal displacement at the target section with the construction process. The horizontal displacement of the main line and ramp hance monitoring points under different lithological grades are shown in Figure 14.

Figure 14a indicates that the deformation curve of the hance of the main line tunnel initially declined and then rose; afterwards, it became stable. The reason was that after the excavation of the upper half step, the section did not form a closed loop, and the lower half step bulged and squeezed the hance to move to the side of the surrounding rocks. After the excavation of the lower half step, the hance moved to the free face in the tunnel, the displacement curve rose, and a closed-loop stable structure was formed. Afterwards, the displacement curve gradually flattened. Under grades III and IV lithologic conditions, the horizontal displacement of the hance was well controlled, and the horizontal displacement of the hance tended to be close to 0. Under the grade V lithologic condition, residual

deformation was observed after the horizontal curve of the tunnel was flattened. Figure 14b shows that the ramp hance displacement curve trend was similar to that of the main line tunnel, but the hance had a large residual deformation. Comparison of the horizontal displacement of the hance after tunnel penetration showed that the order was grade III surrounding rock < grade IV surrounding rock < grade V surrounding rock. The difference between grades III and IV was 0.58 cm, and the difference between grades III and V was 3.27 cm.

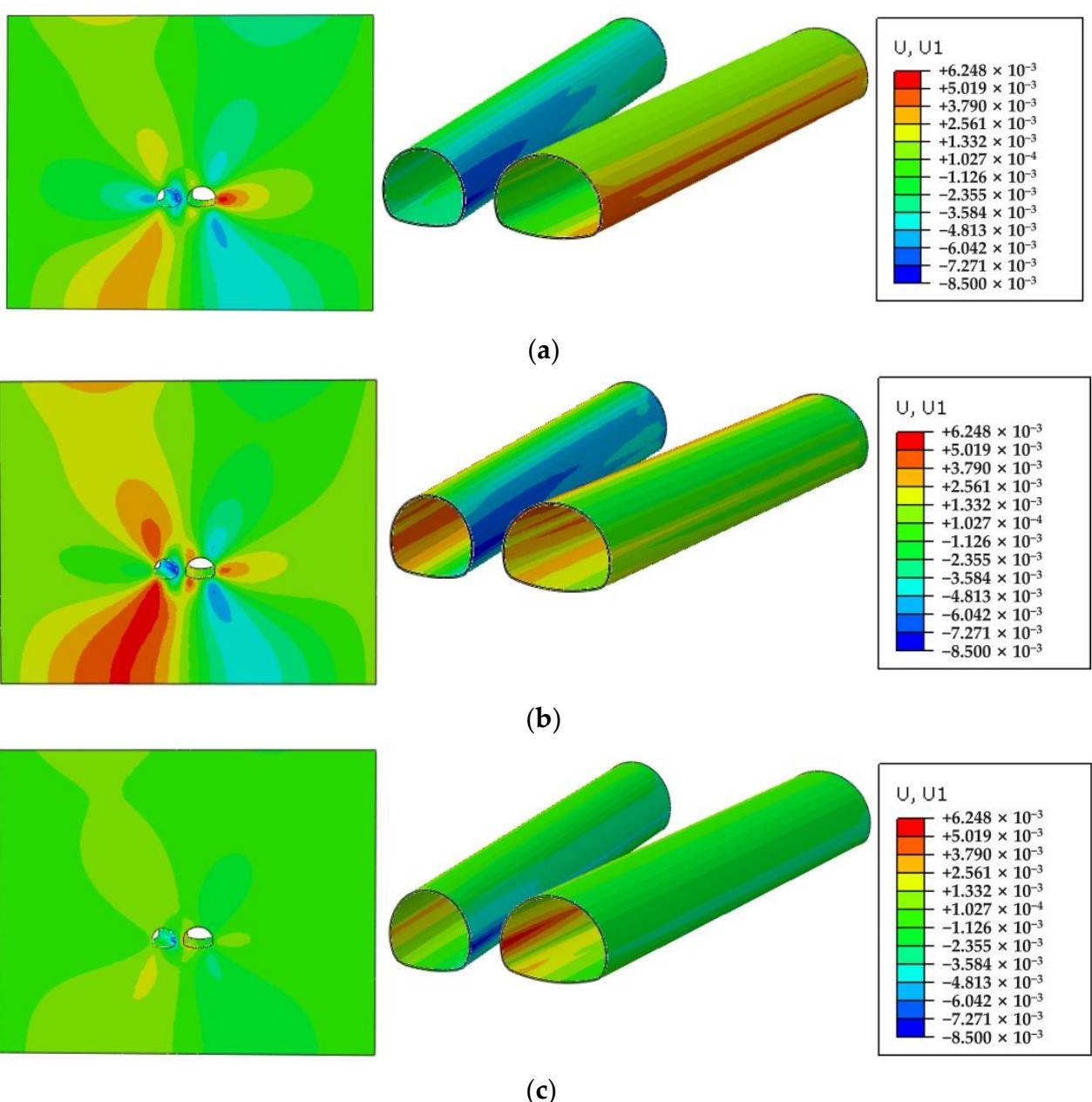

**Figure 13.** Horizontal displacement distribution of surrounding rock and lining after tunnel penetration: (**a**) Grade III rock mass; (**b**) Grade IV rock mass; (**c**) Grade V rock mass.

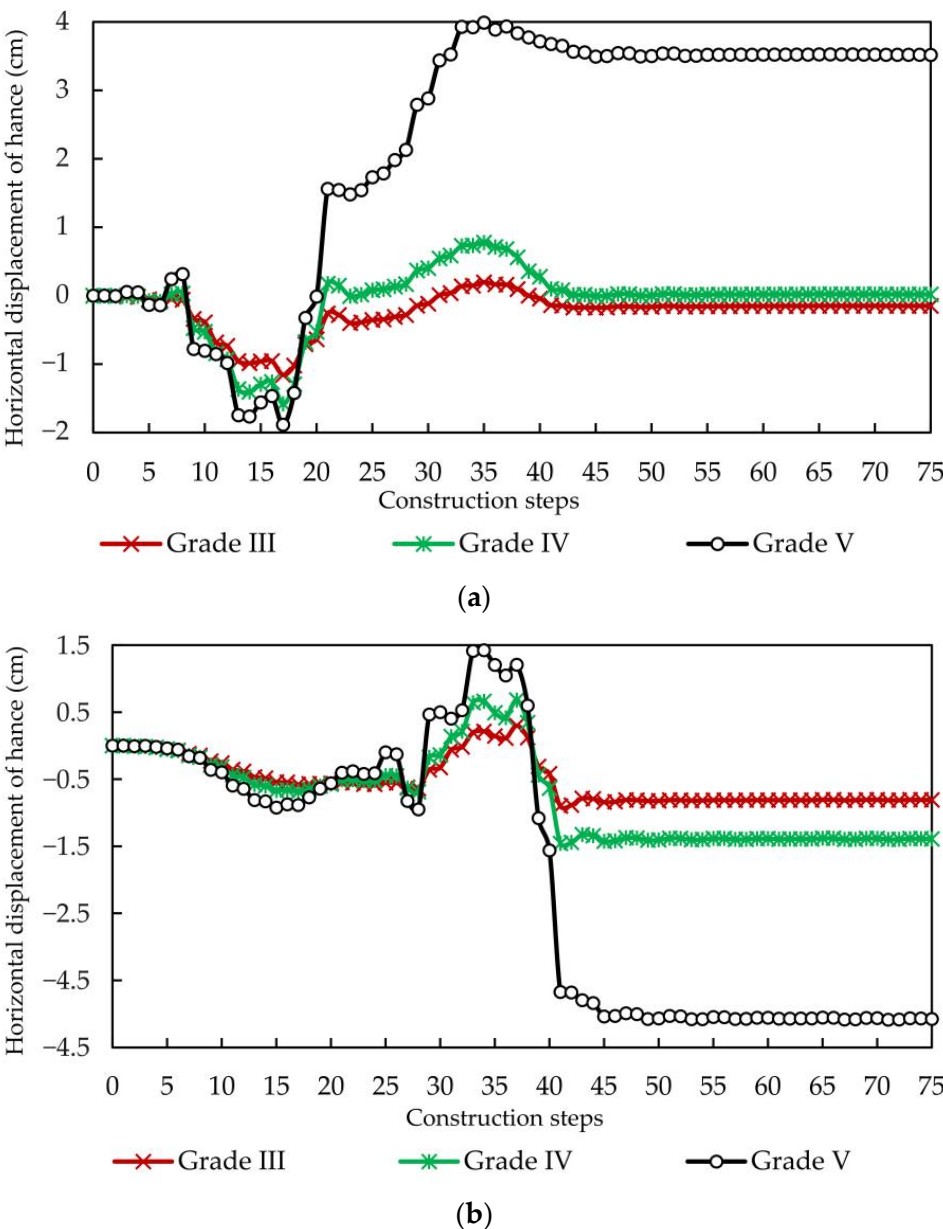

**Figure 14.** Variation curves of hance horizontal displacement under different lithology grades: (**a**) Main line tunnel; (**b**) Ramp.

4.2.2. Stress Analysis of Supporting Structure

Figure 15 presents the maximum principal stress distribution of the main line tunnel and ramp support structure under different lithology grades, and the positive values in the figure represent tension. The main line and ramp vault and invert were tensile areas. A comparison of the maximum and minimum peak principal stresses in different lithological grades showed that the peak tensile stress of grade III surrounding rock was the largest, followed by grade IV surrounding rock and grade V surrounding rock, which had the smallest value. The peak compressive stress of grade III surrounding rock was the smallest and that of grade V surrounding rock was the largest. Given the weak ability of concrete to bear tensile stress, the lining should maintain a certain thickness even under grade III surrounding rock to improve safety; the lining thickness should also be increased under grade V surrounding rock to bear high pressure loads.

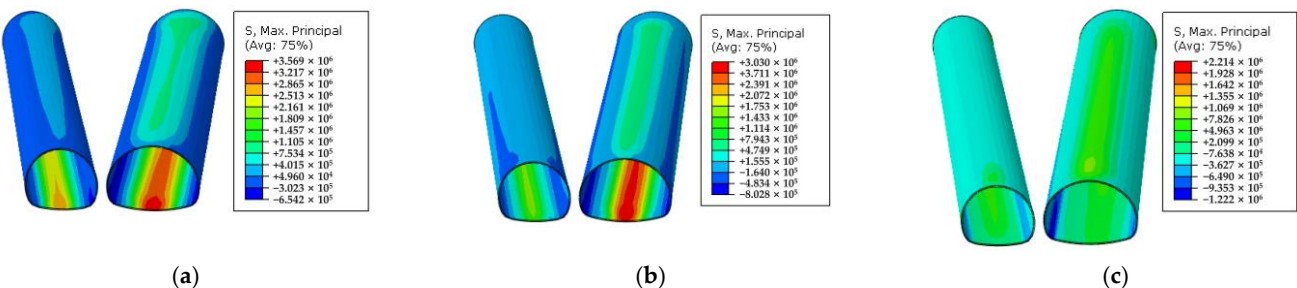

**Figure 15.** Maximum principal stress distribution of main line tunnel and ramp support structures: (**a**) Grade III rock mass; (**b**) Grade IV rock mass; (**c**) Grade V rock mass.

Overall, the vertical displacement of the ground and vault was positively correlated with the elastic modulus of the rock mass, which was consistent with the conclusion obtained by Wang et al. [37] that with the increase of surrounding rock grade, the deformation of the tunnel vault became smaller. The horizontal displacement of the main line tunnel was smaller than that of the ramp under the surrounding rock grades III and IV. The area from the hance to the arch foot in the grade V surrounding rock was prone to large deformation. Attention should be paid to the tension zone at the bottom of the inverted arch under the grade III surrounding rock.

*4.3. Analysis of the Construction Behaviour of Small Clear-Distance Tunnels under Different Footage Lengths*

Footage length is an important factor to weigh construction progress and construction safety. When the surrounding rocks are relatively stable and the section is small, increasing the footage can increase the construction speed and is conducive to the economic efficiency of the project. When the section is special, the footage length needs to be controlled to ensure construction safety. The footage of the small clear-distance tunnel in this study was set to 3, 4, and 5 m. The tunnel was excavated by the step method; the main line tunnel was excavated first. The main line and ramp were staggered by 15 m, and the step face was staggered by 15 m. The surrounding rocks all belonged to grade III. The section 20 m away from the main line tunnel portal was selected as the target section for analysis.

4.3.1. Deformation Analysis of the Surrounding Rocks and Lining

Figure 16 shows the vertical displacement curves of the surface, main line, and ramp vault with different footage lengths at the target section. The vault displacement of the main tunnel was greater than that of the ramp, and the ground settlement was the smallest. The final displacement curves were very close; that is, the final deformation was almost the same under the different footage lengths. In the process of tunnel excavation, the smaller the footage was, the smaller the vertical displacement was. When the footage was 3 m, the displacement curve floated above the curves of 4 and 5 m, and the difference between 4 and 5 m in terms of vertical displacement was small. Specifically, when crossing the section with poor geological and design conditions, the smaller the footage was, the better the settlement control effect was.

To analyze the influence of footage on different sections of the tunnel within a range of 48 m from the tunnel portal, a section of every 6 m was selected for analysis. Figure 17 presents the vertical displacement of the main line tunnel and ramp vault at different distances from the tunnel portal under different footages after the tunnel was penetrated. With the increase in the distance from the portal, the vertical displacement of the main line tunnel and ramp vault decreased. Most of the displacement curve data points with a footage of 3 m were above the curves with footages of 4 and 5 m; that is, the displacement of the vault at the section with a footage of 3 m was smaller than that with footage of 4 and 5 m. This result indicated that reducing the footage length helped control the vertical displacement.

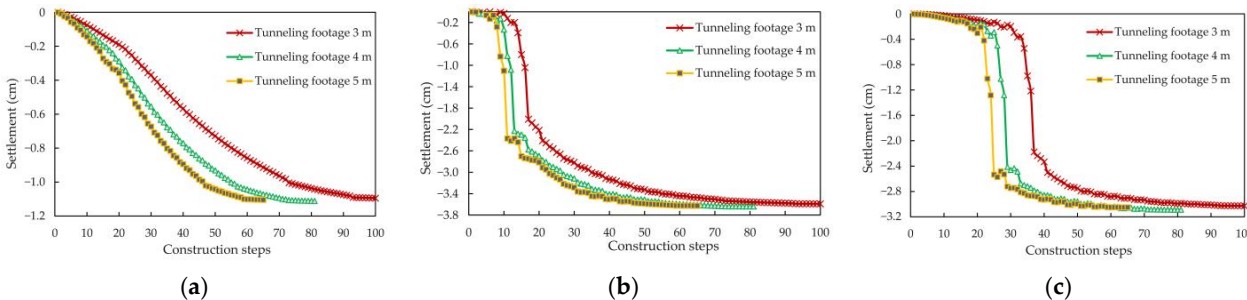

**Figure 16.** The vertical displacement curves of the surface, main line and ramp vault with different footage: (**a**) Surface; (**b**) Main line tunnel; (**c**) Ramp.

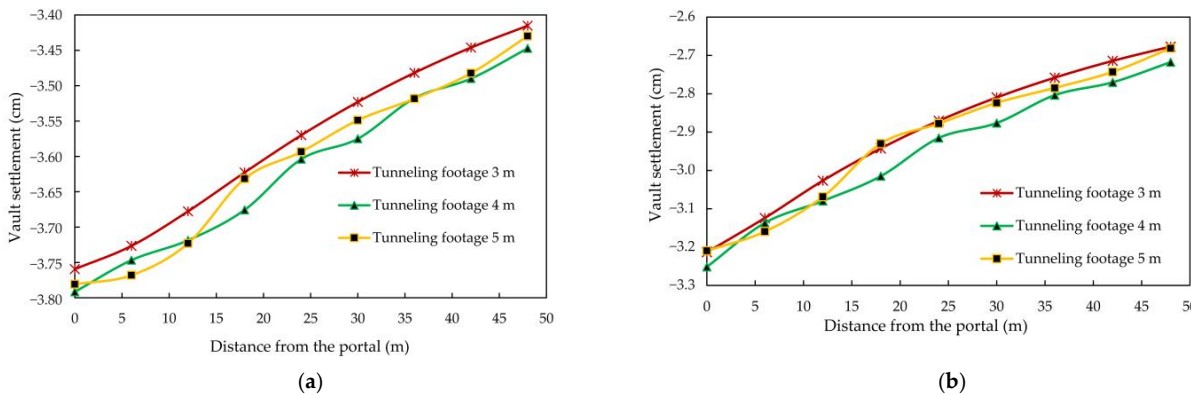

**Figure 17.** The vertical displacement of the main line tunnel and ramp vault at different distances from the tunnel portal: (**a**) Main line tunnel; (**b**) Ramp.

To analyze the characteristics of the horizontal displacement of the tunnel arch waist at different excavation footages, the horizontal displacement of the monitoring point shown in Figure 9 was selected for analysis. Figure 18 shows the variation curve of the horizontal displacement at the main line and ramp hance at different footages. The horizontal displacement of the main line tunnel hance tended to be close to zero after penetration, whereas the ramp tunnel had a tendency to move toward the tunnel because the ramp tunnel was constructed at an 8° inclination, and the redistribution generated by the excavation was offset to the main line tunnel, which had a certain bias effect on the ramp. A small footage had a positive effect on the horizontal displacement of the hance, but it exerted an obvious effect on the deformation of the arch.

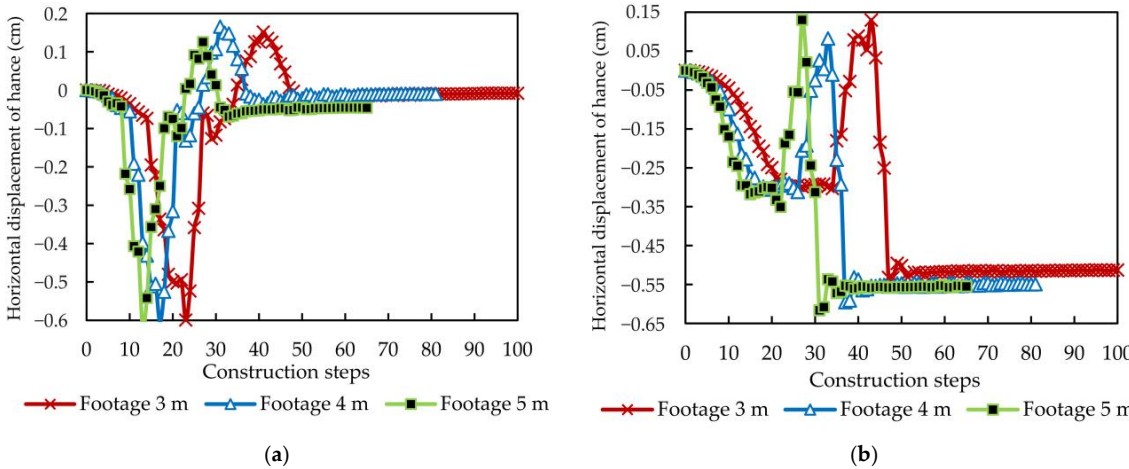

**Figure 18.** Variation curve of the horizontal displacement at the main line and the ramp hance: (**a**) Main line tunnel; (**b**) Ramp.

### 4.3.2. Stress Analysis of the Supporting Structure

Figure 19 shows the distribution of primary lining stress under different footages after the tunnel was penetrated. The maximum stress was distributed in the range of 0–5 m in the left hance of the main tunnel. Under the influence of the overall displacement of the ramp tunnel hance after excavation and support, the horizontal displacement of the main tunnel hance tended to be zero, resulting in a large stress concentration. The stress distribution was consistent in all the footage conditions, but some differences in stress size were observed. The stress of the 4 m footage was the least, followed by that of the 5 m footage. The stress of the 3 m footage was the largest. Therefore, the 4 m footage was beneficial to the stress of the lining.

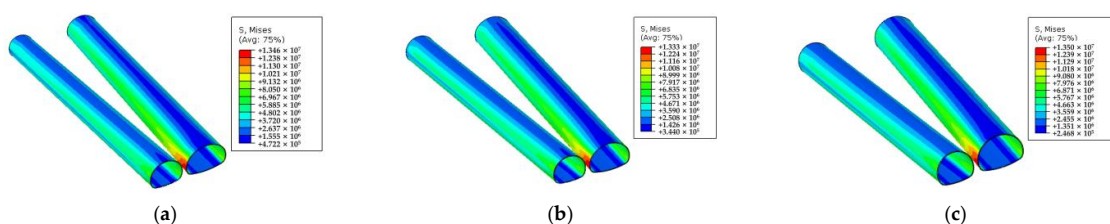

(**a**)          (**b**)          (**c**)

**Figure 19.** Stress distribution of supporting structure in different footage: (**a**) Footage 3 m; (**b**) Footage 4 m; (**c**) Footage 5 m.

Overall, the footage had little influence on the deformation of the surrounding rocks and the stress and deformation of the lining. Reducing the footage was beneficial to the deformation of the vault and hance, but when the footage was small, the overall force was large, and the construction steps increased, which was not conducive to the economic feasibility of the whole project. Therefore, when the geological conditions are good, large footage can be selected for rapid excavation; meanwhile, when crossing special sections, the footage should be reduced to control the deformation and strengthen the support, thereby preventing damage on the support structure.

The excavation effect of small clear-distance tunnels under different excavation schemes, surrounding rock grades, and footage lengths are listed in Table 5. The excavation effect of the double wall heading method in small clear-distance tunnels was better than the other methods, and the same conclusion has been reached by scholars in the study of large section tunnels [35,36]. The higher of the grade of the surrounding rock, the smaller of the deformation and stress caused by excavation, which has been confirmed in literature [37]. Reducing footage was conducive to controlling the deformation and stress of surrounding rock, but not the smaller the better, which is consistent with the literature [38]. See Table 5.

**Table 5.** Excavation effect in different excavation schemes, surrounding rock grades and footage lengths.

| Excavation Effect | Excavation Methods | | | | Lithology Grades | | | Footage Lengths (m) | | |
|---|---|---|---|---|---|---|---|---|---|---|
| | Full-Section Method | Step Method | CD Method | Double Wall Heading Method | III | IV | V | 3 | 4 | 5 |
| Displacement | (1) The influence range of vertical displacement of surrounding rock above the ramp from small to large was the full section, CD, double wall heading and step method. (2) In step and full section methods, the horizontal displacement range of surrounding rock around the tunnel was large. (3) In the CD and double wall heading methods, the overall horizontal displacement was relatively average. (4) In the CD method and double wall heading methods, the overall horizontal displacement of the lining structure was mainly concentrated on the temporary structural deformation. (5) The double wall heading method could maintain the integrity of the primary lining displacement, and the horizontal displacement of the primary lining at the hance was small. | | | | The worse the rock mass quality was and the lower the grade was, the greater the displacement was. | | | The displacement of the vault at the section with footage of 3 m was smaller than that with footage of 4 and 5 m. | | |

**Table 5.** *Cont.*

| Excavation Effect | Excavation Methods | | | | Lithology Grades | | | Footage Lengths (m) | | |
|---|---|---|---|---|---|---|---|---|---|---|
| | Full-Section Method | Step Method | CD Method | Double Wall Heading Method | III | IV | V | 3 | 4 | 5 |
| Stress | The overall stress of the lining was reasonable in the CD and double wall heading methods, but a large range of tensile stress was observed in the full section and step methods. | | | | (1) The peak tensile stress of grade III was the largest, followed by grade IV and grade V. (2) The peak compressive stress of grade III was the smallest, and that of grade V was the largest. | | | The stress of the 4 m footage was the least, followed by that of the 5 m footage. The stress of the 3 m footage was the largest. | | |

## 5. Conclusions

The small clear-distance tunnel of Xiamen Haicang Evacuate Channel Project was selected as background, and different excavation schemes, lithology grades, and footage lengths were employed to analyze the construction mechanical behaviour of a small clear-distance tunnel. The following conclusions were obtained:

(1) For an urban super large and complex underground interchange hub, main line, and ramps were all constructed underground. Complex adjacent tunnels form complex small clear-distance tunnels whose construction mechanical behaviour of the small clear-distance tunnels was vital for the underground engineering system.

(2) When the CD method was used for the excavation of the small clear-distance tunnel, the vertical displacement of the main line and ramp tunnel vaults was the smallest. The double wall heading method could effectively control the horizontal displacement of the tunnel arch waist. The overall stress state of the tunnel lining in the CD and double sidewall pilot pit methods was reasonable, whereas the tunnel lining in the full section and step methods had a large range of tensile stress.

(3) In the CD method, the vertical displacement of the main line and ramp tunnel vault of the small clear-distance tunnel was the smallest. The double wall heading method could effectively control the horizontal displacement of the tunnel hance. The overall stress of the tunnel lining in the CD and double wall heading methods was reasonable, but a large range of tensile stress in the tunnel lining was observed in the full section and step methods.

(4) The vertical displacement of the surface, main line tunnel vault, and ramp vault increased with the increase in the elastic modulus of the rock mass. In the rock mass with grades III and IV, the horizontal displacement of the main line tunnel hance was smaller than that of the ramp, and the area from the hance to the arch foot was prone to large deformation when no temporary support was available in the grade V rock mass.

(5) Compared with the excavation scheme and lithology grade, the excavation footage length of the small clear-distance tunnel had less influence on the surrounding rock deformation and lining stress/deformation.

(6) Reducing the tunnel excavation footage was conducive to reducing the deformation of the vault and hance. Therefore, when the geological conditions are good, a large footage can be selected for rapid excavation. Meanwhile, when crossing special sections, the footage should be reduced to control deformation.

**Author Contributions:** Conceptualization, J.W.; methodology, J.W. and Z.W.; software, Z.W.; validation, L.L., Z.L., J.W., A.C., H.L., Y.S. and X.L.; formal analysis, J.W. and Z.W.; investigation, J.W., X.L., A.C., H.L. and Y.S.; resources, J.W.; data curation, Z.W.; writing—original draft preparation, J.W., Z.W. and A.C.; writing—review and editing, J.W. and A.C.; project administration, L.L. and Z.L. All authors have read and agreed to the published version of the manuscript.

**Funding:** This research was funded by the Shanghai Municipal Science and Technology Project (18DZ1201301; 19DZ1200900); Xiamen Road and Bridge Group (XM2017-TZ0151; XM2017-TZ0117); the project of Key Laboratory of Impact and Safety Engineering (Ningbo University), Ministry of Education (CJ202101); Shanghai Municipal Science and Technology Major Project (2021SHZDZX0100) and the Fundamental Research Funds for the Central Universities; Key La-boratory of Land Subsi-dence Monitoring and Prevention, Ministry of Natural Resources of the People's Republic of China (No. KLLSMP202101; KLLSMP202201); Suzhou Rail Transit Line 1 Co., Ltd. (SURT01YJ1S10002); China Railway 15 Bureau Group Co., Ltd. (CR15CG-XLDYH7-2019-GC01).

**Institutional Review Board Statement:** The study did not require ethical approval.

**Informed Consent Statement:** The study did not involve humans.

**Data Availability Statement:** The data presented in this study are available on request from the corresponding author.

**Conflicts of Interest:** The authors declare no conflict of interest.

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
