# Peer review of "Mechanical Behavior and Excavation Optimization of a Small Clear-Distance Tunnel in an Urban Super Large and Complex Underground Interchange Hub"

_applsci, doi:10.3390/app13010254_

Round 1
Reviewer 1 Report
In this manuscript, taking the small clear distance tunnel of Xiamen Haicang Evacuate-channel as the research object, FLAC3D was used to analyze the influence of different excavation schemes, lithological grades and footage lengths on tunnel stability. The deformation and stress characteristics of surrounding rock and lining structure in the excavation schemes of the full section method, step method, CD method and double wall heading method, in the grade of â…¢, â…£ and â…¤ rock mass, and in the footage length of 3 m, 4 m and 5 m were introduced. The paper has novelty and advantages for this field research work, but needs further modification. Hence, major revision is recommended.
Some specific comments are listed as follows:
1. English writing is generally good, some tenses need further confirmation.
2. The clear distance of the small clear distance tunnel needs to be clarified.
3. There is an error in Figure 2, please modify.
4. In 150-160 lines, this paragraph is redundant.
5. In 166-168 lines, “According to the influence range of the excavation, stress is 3-5 times of the tunnel radius, the boundary dimensions of the model were 200 m in the X direction, 110 m in the Z direction, and 160 m in the Y direction.” Please rewrite this sentence.
6. Please further explain why the influence range from small to large is the full section method < CD method, double wall heading method < step method.
7. Figures 6(c) and 8(c) are obviously different from other figures, are there any errors?
Reviewer 2 Report
Dear Authors
Originality of the research is good, you can check typo errors once again in your manuscript, manuscript content is easily followed by the researchers, it can be accepted for publication as it is
Reviewer 3 Report
I think that this manuscript can be accepted for its publication.
Author Response
Thank you very much.
Reviewer 4 Report
Peer Review Report of the manuscript number applsci-2112904:
The modeling of a small clear-distance tunnel is very interesting.
Intensive English language Editing is required for the text of the manuscript. For instance, a typo in word “adopted” was noticed in line16 in the abstract.
The significance of this study has to be supported when the researchers support their statement from other most recent references.
What is the acronym of CD method stands for? What software was used in section 3 line 168?
what is the boundary condition applied? this must be metioned and showed in a Figure.
what are the properties of the C3D8R element? more details about this element must be provided.
Based on Figure 3 and section 3 in line 168, Why you did not use the symmetry feature in modeling the model, so that the number of elements can be reduced to 75%? This needs to be justified from the literature.
What is the reference of the material parameters in Table 2? What is the compressive and/or tension strength of these materials?
Can the model predict the effect of ground water on the tunnel?
It is required to have a table before the conclusion section to summarize the findings of the methods used in the numerical model and compare that with findings of similar methods from another literature. Not necessarily to be on the same type of tunnel, but at least to highlight the significance of this study.
54%
of the cited references are old, this has to be addressed. The number of the cited references require to be increased , especially in section 4 in line 199 to support the discussion from the literature
